# Endophyte-Mediated Stress Tolerance in Plants: A Sustainable Strategy to Enhance Resilience and Assist Crop Improvement

**DOI:** 10.3390/cells11203292

**Published:** 2022-10-19

**Authors:** Muhammad Kamran, Qari Muhammad Imran, Muhammad Bilal Ahmed, Noreen Falak, Amna Khatoon, Byung-Wook Yun

**Affiliations:** 1School of Molecular Sciences, The University of Western Australia, M310, 35 Stirling Hwy, Perth, WA 6009, Australia; 2Department of Medical Biochemistry & Biophysics, Umea University, 90187 Umea, Sweden; 3Laboratory of Plant Molecular Pathology and Functional Genomics, Division of Plant Biosciences, College of Agriculture and & Life Science, Kyungpook National University, Daegu 41566, Korea; 4BK21 FOUR KNU Creative BioResearch Group, School of Life Sciences, College of Natural Sciences, Kyungpook National University, Daegu 41566, Korea; 5Department of Botany, Kohat University of Science and Technology, Kohat 26000, Pakistan

**Keywords:** endophytes, plant defense, drought, salinity, temperature, crop improvement

## Abstract

Biotic and abiotic stresses severely affect agriculture by affecting crop productivity, soil fertility, and health. These stresses may have significant financial repercussions, necessitating a practical, cost-effective, and ecologically friendly approach to lessen their negative impacts on plants. Several agrochemicals, such as fertilizers, pesticides, and insecticides, are used to improve plant health and protection; however, these chemical supplements have serious implications for human health. Plants being sessile cannot move or escape to avoid stress. Therefore, they have evolved to develop highly beneficial interactions with endophytes. The targeted use of beneficial plant endophytes and their role in combating biotic and abiotic stresses are gaining attention. Therefore, it is important to experimentally validate these interactions and determine how they affect plant fitness. This review highlights research that sheds light on how endophytes help plants tolerate biotic and abiotic stresses through plant–symbiont and plant–microbiota interactions. There is a great need to focus research efforts on this vital area to achieve a system-level understanding of plant–microbe interactions that occur naturally.

## 1. Major Challenges to Plant’s Growth and Production

The world needs more food to feed the growing population. It is a harsh reality that the world population is increasing rapidly, with an increase of 81 million people (1.05%) every year [1]. The growth is more severe in the developing world, such as Pakistan, where the annual increase is 2.0% [2]. The population spiked in 1962–63, when the world population grew at a rate of 2.2% [3], alerting agronomists and plant scientists to accommodate the demand. Thanks to science, the development of high-stress-resistant new varieties and plant breeding-based enhanced production has increased global food production. According to an estimate by the Food and Agriculture Organization (FAO) [4], primary crop production increased by almost 50% compared with 2000, whereas vegetable oil production increased by 108%. However, this increase in food production comes with a cost. According to the same report, pesticide use went up by one-third between 2000 and 2018, and the use of nitrogen fertilizers increased by 58%. In quantitative terms, 190 million tons of fertilizer were added to the soil during this period [5]. Since then, the uses of fertilizers and other agrochemicals have increased, as they are known to increase yield without much effort [6]. According to research, these agrochemicals have severe consequences on human health, including immune system disorders, hormonal problems, kidney disorders, reproductive problems, and cancer [7]. A detailed account of these agrochemicals and their impact on human health was provided by Srivastav [8].

Therefore, sustainable agriculture is key to coping with increasing food demand, with less detrimental or no effects on human health. In this context, plant microbiomes play a vital role in plant growth and maintenance of soil fertility for sustainable agriculture/food production. Soil microbes are important regulators of global nutrient balance in ecosystems [9]. Plant and microbial associations, such as rhizospheric, epiphytic, and endophytic associations with growth-promoting properties, are critical for sustainable agriculture. These microbes, either directly in the form of phytohormones or indirectly, such as nitrogen fixation, can promote plant growth and therefore can be a reasonable alternative to chemical fertilizers. To that end, this review focuses on the core concerns surrounding the potential for developing innovative microbial-assisted selection of plants, maximizing rhizosphere/root microbiome beneficial connections, with a special focus on endophytes.

## 2. Endophytes: Friends or Foes?

Endophytes are microorganisms that inhabit plants. Endophytes are known to promote the growth of host plants in several ways, including detoxification of toxic compounds, defense against pathogens, and production of plant growth-promoting hormones [10]. Endophytes are also known to produce biotechnologically important metabolites having anticancer and antimicrobial properties [9]. They also help in the recycling of nutrients and act as bio transformers of different chemicals [9]. Endophytes are diverse and relatively less explored; however, new endophytes with diverse functions are being added to the list with the development of research on plant-microbe interactions.

### Colonization and Distribution

Endophytes can be bacterial or fungal. Bacterial endophytes are a class of endosymbiotic microorganisms that live in intercellular or intracellular compartments, apoplasts, water-conducting xylem vessels of shoots and roots, intercellular spaces of cell walls, flowers [11], fruits [12], and seeds [13]. Unlike bacterial phytopathogens, endophytes do not cause disease or morphological abnormalities in plants [14]. The number of bacterial endophytes in plants varies across species, and the type of tissue ranges from 100 to 9 × 10^9^ bacteria per gram of plant tissue [15]. Generally, endophyte density is higher in roots and belowground parts than in aboveground parts [14]. However, this number may vary, as migration of endophytes from roots to leaves has been demonstrated in rice [15], suggesting that endophytes enter the host plant through the roots [14]. Although they live inside plants, the population density and distribution of bacterial endophytes are influenced by both the biotic and abiotic factors faced by plants, including microbe–microbe interactions, plant–microbe interactions, and environmental stresses [16].

Over a century of research suggests that in natural ecosystems, most plants are symbiotic with mycorrhizal fungi or fungal endophytes [17]. This fungal partner can profoundly impact plant evolution, fitness, and distribution [18], defining plant communities [19]. Plant and fungal mycorrhizal and endophytic associations date back more than 400 million years as per fossil records [20,21] and were likely associated at the time of initial footprints of plants on land, thus playing a key role in the evolution of life on land [22]. Similar to their bacterial counterparts, fungal endophytes reside entirely within the plant host, where they may grow within underground or aboveground parts and sporulate during plant or host tissue senescence [23]. Fungal endophytes are classified into four classes based on their host range and mode of transmission [22]. Class 1 endophytes represent Clavicipitaceous fungi, characterized by both vertical and horizontal transmission with their seeds, and can be found in some grasses in cold habitats, while the remaining three classes are non-claviciptaceous endophytes. Class 2 endophytes represent endophytes that grow in both above- and below-ground tissues. These can transmit vertically and horizontally and can be recovered from asymptomatic tissues of non-vascular plants, ferns, conifers, and some angiosperms. Class 3 endophytes generally have a wide host range and colonize only the upper plant parts, and their mode of transmission is horizontal. Class 4 endophytes colonize roots and are transmitted horizontally.

## 3. Endophytes Modulate Plant Defense Responses

Plants generally respond to pathogenic infections, either by pathogen-associated molecular pattern (PAMPs)-triggered immunity (PTI) or effector-triggered immunity (ETI). A detailed explanation of how plants prioritize resistance responses was given by Jones and Dangl [24]. The fundamental question is how endophytes overcome/avoid such resistance responses by plants while infecting their host. To answer this question, we need to understand the mechanism of inconspicuous entry of endophytes into the host plant and their management to thrive there through a mutually symbiotic relationship.

Some endophytes are seed-borne, and they are present in germinated plants [25]. In plants with vegetative propagation, endophyte colonization is relatively easier because they can transmit their endophytic microbiota to the next generation. In others, it starts with root exudates, which are rich in biomolecules and chemo-attractants such as flavonoids, sugars, amino acids, organic acids, and phenolic compounds. Root exudates are also rich in nutrients and water and therefore attract all types of other microbes [26]. Effective endophyte colonization entails compatible plant–microbe interactions. Potential points of entry for endophytes are cracks formed at the emergence of lateral roots or zones of root elongation. To make an entry, some bacteria must find their way to these apertures for successful colonization. Dong et al. [27] showed that *Klebsiella* sp/strain Kp342 colonizes the lateral root junctions in wheat and alfalfa. Similarly, *Herbaspirillum seropedicae* and *Gluconacetobacter diazotrophicus* dominate colonization at later-root junctions [28]. Some endophytes enter through infection colonization, where cellulolytic and pectinolytic enzymes produced by endophytes come into play [14], such as pectate lyase, which has been implicated in the colonization of *Klebsiella* strains [29]. Figure 1 summarizes the potential entry and colonization of endophytes and their interactions with the host plant tissues. For entry into the host plant, fungal endophytes produce chitin deacetylase enzymes that deacetylate chitosan oligomers and are camouflaged by plant pattern recognition receptors (PRRs) [30]. Similarly, other evidence suggests that some endophytic bacteria release their microbe-associated molecular patterns (MAMPs), which either leads to misrecognition by plant PRRs or induces a comparatively weak and transient defense response [31]. Similarly, cell wall–degrading enzymes such as endoglucanases and polygalacturonase are used by *Burkholderia* sp. for infection in *Vitis vinifera* [11].

Endophytes help in protection against pathogens by being directly involved in the production and release of secondary metabolites or compounds with antimicrobial properties, such as siderophores, antibiotics, and hydrolytic enzymes, which can help in culminating or reducing the invading pathogens or through indirect mechanisms by competing with the pathogen for space and available nutrients [32]. Endophytes are known to play a role in systemic acquired resistance (SAR) against plant pathogens [33]. The phytohormones salicylic acid (SA) and jasmonic acid (JA) are vital for plant defense responses. Some endophytes have increased levels of SA and JA, thereby triggering plant defense responses against pathogens. For example, Waqas et al. [34] reported that the gibberellin-producing endophytes *Penicillium citrinum* strain LWL4 and *Aspergillus terreus* strain LWL5 trigger SAR against stem rot in sunflowers by inducing the SA and JA pathways. Similarly, *Fusarium solani* induces SAR against *Septoria lycopersici* by inducing pathogenesis-related (PR) gene expression [35]. Table 1 summarizes a list of some of the endophytes that are reported to have defensive role against phyto-pathogens. Foliar application of endophytes also affects plant defense pathways [36,37,38]. For example, foliar application of the endophytic fungus *Colletotrichum tropicale* reduced *Phytophthora* infection in the Cacao tree (*Theobroma cacao)* [39]. The reduction in *Phytophthora* infection could be attributed to the induction of defense-related pathways, as inoculation with *C. tropicale* triggered several components of the ethylene (ET) defense pathway and induced several disease-related genes in *T. cacao* and *Arabidopsis thaliana* [40,41].

Endophytic bacteria are well known for producing compounds with antimicrobial properties, such as lipopeptides. The most studied lipopeptides are those isolated from *Bacillus* and *Paenibacillus* species [42]. Similarly, another endosymbiont, *Pseudomonas viridiflava*, was also reported to produce ecomycins, a lipopeptide containing unique amino acids, such as β-hydroxy aspartic acid and homoserine [43]. Endophytic *Pseudomonas putida* strain BP25 inhibits several phytopathogens, including *Pythium myriotylum*, *Gibberella moniliformis*, *Phytophthora capsica*, *Thizoctonia solani*, *Colletotrichum gloeosporioides*, and the plant-parasitic nematode *Radopholus similis* through the production of several volatile substances [44]. Similarly, *Rhizobium* is known for siderophore production and is used as a biocontrol agent for pathogenic *Macrophomina phaseolina,* which causes charcoal rot in several crops [45]. In addition to lipopeptides, several other endophytes have been reported to produce other antimicrobial compounds, such as mycosubtilin, surfactin, lichenysin, bacillomycin, iturin, fengycin, and plipastaitin [46]. A detailed list of different antimicrobial compounds produced by various endophytes was reviewed in ref. [47].

Plants also produce some low molecular weight antimicrobial molecules known as phytoalexins, including some other groups of secondary metabolites, such as terpenoids [48]. Endophytes can also increase the production of secondary metabolites. Reports have suggested an increase in terpenoid content by the endophytic fungus *Fusarium* spp. in *Euphorbia pekinensis* [49]. Similarly, plants inhabited by endophytes have been reported to exhibit increased cellulose content, lamina density, and leaf stiffness, thereby reducing herbivory rates, particularly against leaf-cutting ants [36,37]. Thus, resistance against phytopathogens is achieved by eliciting host responses or producing specific metabolites by endophytes working in close coordination to cope with pathogenic ingress.

**Table 1 cells-11-03292-t001:** List of endophytes associated with defense against various plant pathogens.

Endophyte	Host Plant	Pathogen	Target Pathway	Reference
*Bacillus* spp.	*Oryza sativa*	*Pyricularia oryzae*	Induce systemic resistance	[50]
*Rhizobium etli*	*Phaseolus vulgaris*	*Pseudomonas syringae pv.phaseolicola*	Callose deposition, SA, and JA-dependent gene induction	[50]
*Bacillus* spp.	*Nicotiana tabacum*	*Pseudomonas syringae pv.tobaco*	Induce systemic resistance	[50]
*Paenibacillus*	*Triticum aestivum*	*Mycosphaerella graminicola*	Defense pathway	[51]
Bacillus spp.	*Oryza sativa*	*Pyricularia oryzae*	Antioxidant defense activities	[52]
*Phomopis cassiae*	*Cassia spectabilis*	*Cladosporium sphaerospermum*	unknown	[53]
*Streptomyces* strain, DEF09	*Triticum aestivum*	*Fusarium graminearum*	Chitinase production	[54]
*Daldinia eschscholtzii*	*Helianthus tuberosus, Zingiber officinale, Stemona tuberosa*	*Colletotrichum acutatum, Sclerotium rolfsii*	Production of antimicrobial compounds	[55]
*Paraburkholderia*	*Beta vulgaris*	*Rhizoctonia solani*	Systemic acquired resistance	[56]
*Penicillium citrinum*	*Helianthus annus L*	*Sclerotium rolfsii*	SA, JA pathway	[34]
*Fusarium solani*	*Solanum lycopersicum*	*Septoria lycopersici*	SA-dependent PR gene pathway	[35]
*Paenibacillus* sp. strain B2	*Triticum aestivum*	*Zymoseptoria tritici*	Induce flavonoid and phytohormonal pathways	[57]
*Paraconiothyrium* sp.	*Fraxinus excelsior*	*Hymenoscyphus fraxineus*	Unknown	[58]
*Cladosporium* spp.	*Pinus monticola*	*Cronartium Ribicola*	Induced resistance	[59]
*Venturia fraxini*	*Fraxinus excelsior*	*Hymenoscyphus fraxineus*	Production of antifungal compounds	[60]
*Flavobacterium*	*Beta vulgaris*	*Rhizoctonia solani*	Chitinase, nonribosomal peptide synthetases (NRPSs), and polyketide synthases (PKSc)	[61]
*Gluconacetobacter diazotrophicus*	*Arabidopsis thaliana*	*Ralstonia solanacearum*	Callose deposition	[62]
*Piriformospora indica*	*A. thaliana*	*Golovinomyces orontii*	JA-dependent defense pathway	[63]
*Pseudomonas simiae*	*A. thaliana*	*Mamestra brassicae*	JA and ET	[64]
*Bacillus pumilus* strain	*A. thaliana*	Cucumber mosaic virus	Ethylene pathway	[65]
*Bacillus subtilis*	*A. thaliana*	*P. syringae* pv. Tomato DC3000	SA, ET pathway	[66]
*Bacillus aryabhattai*	*Nicotiana tabacum, A. thaliana*	*Botrytus cinerea*	SA, JA	[67]

## 4. Endophytes and Abiotic Stresses

Abiotic stress is a rising challenge in crop production worldwide. The most common abiotic stressors include temperature, drought, salinity, and nutrient deficiency (Figure 2). These stresses may cause alterations in metabolomics and transcriptomics, which change the exudates from the roots and leaves and, in turn, influence the microbial community associated with plants [68]. In the soil ecosystem, plant-associated microbes play a central role, working as natural partners that facilitate local and systemic mechanisms in plants to defend against adverse environmental conditions. This section highlights the underlying mechanisms and how plant–microbe interactions modulate responses under major abiotic stresses, such as temperature, drought, salinity, and nutrition stress.

### 4.1. Temperature

Temperature is the most common stress experienced by plants. The average global temperature and the frequency of extreme weather events have increased considerably over the past several decades owing to climate change. According to the results of several climate model simulations, the average temperature of the world is predicted to be between 1.1 and 5.4 °C warmer in 2100 than it is today [69]. The main effects of temperature stress include alterations in plasma membrane activity, transpiration, reduced photosynthesis, enzymatic dysfunction, cell division, plant growth, and development [70]. Previous studies have shown that plant response to heat stress may be influenced by microbe interactions [71,72,73].

Reports suggest that the endophytic bacteria *Burkholderia phytofirmans* strain PsJN can alter photosynthesis and the carbohydrate metabolism involved in chilling stress, thereby enhancing the cold tolerance of grapevine plants [11,74]. Bacterial endophytes induce adaptation to chilling, resulting in reduced cell damage, increased photosynthesis, and accumulation of chilling stress-associated metabolites, such as proline, starch, and phenolic compounds [14].

Some microbes can protect themselves from extreme temperature conditions by enhancing the production of heat- and cold-tolerant proteins. Molecular chaperones have been reported to play essential roles in heat stress tolerance in microbes [75,76]. For example, the *DnaK* gene encoding heat shock protein (Hsp70), a well-known chaperone in heat stress tolerance, was found to be highly expressed in *Alicyclobacillus acidocaldarius* during heat and cold stress [77]. Microbes adapted to low or high temperatures could alleviate the negative impacts of extreme temperatures and expand the host plant’s ability to survive these conditions. Khan et al. reported that thermotolerant *Bacillus cereus* strain SA1 alleviated heat stress in soybean plants by enhancing antioxidant defense enzymes and altering hormonal profiles [78]. Several other studies have reported a similar mechanism of heat stress tolerance in host plants mediated by thermotolerant endophytes and rhizobacteria, such as *Bacillus amyloliquefaciens* in rice [79], *Pseudomonas putida* strain AKMP7 in wheat [80], and *Pseudomonas* sp. strain AKMP6 in sorghum [81]. In some cases, the relationship between plants and microbes is important for both partners. The symbiosis between the fungus *Curvularia protuberata* and panic grass is interesting, as this relationship allows both organisms to survive high soil temperatures. However, neither the host plant nor the fungus can individually tolerate high-temperature conditions [82]. Furthermore, *C. protuberata* is required to be infected by *Curvularia,* a thermal-tolerant virus (CThTV), to confer heat tolerance to the host plant [82]. Moreover, *C. protuberate*-mediated heat tolerance has been observed in tomatoes [83], indicating that the underlying process may have a wide range of applications to help plants survive at elevated temperatures [84].

### 4.2. Salinity

Salinity is a global issue that affects agricultural productivity. The regions affected by salinity are gradually expanding owing to insufficient precipitation, poor irrigation systems, salt deposition, water pollution, and other environmental factors [85]. Although the development of salt-tolerant crops has been an important scientific aim, there has not been much progress in this direction because not many of the key genetic traits that affect salt tolerance have been identified [86,87,88]. Introducing salt-tolerant microorganisms that promote plant growth is a potential alternative to increase crop tolerance to salinity stress [89].

Plants exhibit a two-stage response to salinity stress. During the first phase, growth slows drastically, and stomata close in response to decreased water potential. Sodium ions are built up in the second phase, increasing oxidative stress, which in turn damages photosynthetic components [87]. In addition to harmful stress by-products, reactive oxygen species (ROS) may also function as signal molecules to initiate defense mechanisms against salt and other stressors [90].

Multiple mechanisms have been implicated in endophyte-mediated salinity tolerance. One approach is to produce large quantities of organic osmolytes such as trehalose, proline, glycine, and betaine, which offer an adaptive response to salinity stress [85]. Some endophytes help plants tolerate salinity stress by reprogramming the antioxidant defense system and hormonal profiles. For example, Khan et al. [91] reported that bacterial endophytes *Bacillus cereus, Micrococcus yunnanensis, Enterobacter tabaci, Curtobacterium oceanosedimentum, Curtobacterium luteum,* and *Enterobacter ludwigii* isolated from halotolerant plants enhanced the salt stress tolerance of rice by altering defense enzymes, indole-3-acetic acid, abscisic acid (ABA), and gibberellin levels. Similar mechanisms have been observed for soybean [92], tomato [93], cucumber [94], and maize [95].

Some endophytes enhance the ability of the host plant to alleviate salt stress by improving key transcripts that are required for salinity tolerance. For example, the bacterial endophyte *Bacillus subtilis* has been reported to enhance salinity tolerance in *Arabidopsis* by simultaneously downregulating high-affinity K^+^ transporter 1 (HKT1) expression in roots and upregulating it in shoots. The induction of HKT1-dependent shoot-to-root recirculation was likely to result in a 50% reduction in plant-wide Na accumulation [96]. Another study reported that auxins produced by *Bacillus amyloliquefaciens* alleviate salt stress by upregulating the expression of ethylene-responsive element binding protein (EREBP), betaine aldehyde dehydrogenase (BADH), salt overly sensitive 1 (SOS1), and catalase genes in rice [97]. Another endophytic bacterium, *Pseudomonas pseudoalcaligenes,* was shown to enhance salt tolerance in rice by accumulating high concentrations of glycine betaine-like compounds [98]. Some endophytes secrete exopolysaccharides that bind cations in the roots (particularly Na^+^) and inhibit their transport to leaves, thereby reducing salt stress [97]. They proposed that roots from inoculated seedlings had a larger proportion of soil sheaths covering them, which could slow down the passage of apoplastic Na^+^ into the stele.

### 4.3. Drought

Without water, life on Earth is not possible. Too little water (drought) or too much water (flood) can have a considerable influence on several aspects of plant and microbial life. A drought is described as a period when there is insufficient water available for an organism or environment to function at its best [99,100,101]. Drought stress may occur at any plant developmental stage and can affect growth and development to varying degrees based on the onset time, duration, and severity. At the reproductive stage of the plant, drought stress may directly reduce more than 50% of the yield [79,102]. Plant breeders have made remarkable progress in creating drought-tolerant crops, but despite this, they are still unable to satisfy the needs of food security in the face of a rising global population, climate change, and water scarcity [101]. Traditional breeding, however, requires much work from the breeder, takes a long time, and may result in the loss of other desirable features [103,104]. Endophytic microorganisms have been reported to alter plant responses to drought and can be used as an alternative and rapid way to enhance crop productivity.

Plants alter phytohormone abscisic acid (ABA) production in response to drought stress. When ABA concentration rises, it sets off a chain reaction that causes the plant to undergo extensive physiological and transcriptional changes, such as the closing of stomata to decrease water loss via transpiration [105]. Furthermore, cell size and membrane integrity are reduced, ROS are produced, and leaf senescence is promoted, all of which contribute to a reduction in agricultural productivity [106].

Under water-deficient conditions, endophytic microorganisms colonize the rhizosphere and promote plant growth and development through a variety of direct and indirect mechanisms. Plant endophytes can produce plant hormones that promote plant growth and development. Mahest et al. reported that the endophytic actinobacteria *S. olivaceus* strain DE10 enhanced wheat growth via auxin production under drought conditions [107]. *Azospirillum brasilense* improves drought tolerance in *Arabidopsis thaliana* by increasing ABA levels [108]. Endophyte-facilitated drought tolerance via alteration of hormonal profiles has been reported in soybean [109], wheat [110], *Lavandula dentata* [111], cucumber [112], sugar cane [113], and soybean [114].

Additionally, in certain instances, *Rhizophagus irregularis* reduced drought stress in tomatoes and lettuce by colonizing the roots [115]. Interestingly, both drought and colonization by *R. irregularis* increased the synthesis of the phytohormone strigolactone. Since strigolactones are known as host identification signals for arbuscular mycorrhiza (AM) fungi, this implies that they operate as a “call for help” signal and start a positive feedback loop of *R. irregularis* colonization that increases host plant tolerance to drought conditions [116]. A similar positive effect of bacterial endophytes has also been reported in wheat, where endophytes played a key role in metabolite balance and the reduced effects of drought stress [117].

Some endophytes help the host plant survive drought-induced oxidative stress by enhancing antioxidant defense enzymes. Moghaddam et al. [118] reported that endophytic fungi *Neocamarosporium*
*goegapense, Neocamarosporium chichastianum*, and *Periconia macrospinosa* isolated from a desert plant alleviated drought stress in wheat and cucumber by boosting antioxidant defense enzymes. The endophytic bacteria *Bacillus*
*safensis* and *Ochrobactrum pseudogregnonense* increased antioxidant activity and enhanced wheat growth under drought conditions [119]. Similar mechanisms of endophyte-mediated drought tolerance have been reported in maize [120,121], wheat [110], Chinese cabbage [122], and other popular crops [123].

Drought also affects the microbial diversity in the host plant rhizosphere. Xu et al. [124] showed that drought drastically lowered bacterial diversity in the rhizosphere and root endosphere, whereas the bacterial community diversity in the surrounding soil remained mostly unchanged. Furthermore, the authors reported that drought enhanced the abundance of actinobacteria, which in turn resulted in the enrichment of metabolites required for drought tolerance. However, it is still unclear how these drought-enhanced metabolites reconfigure the root microbiome makeup to improve plant stress responses. Nonetheless, this intriguing association raises the possibility that plants, and their associated microbiomes engage in molecular dialogues during drought to change the root microbiota to better tolerate drought stress. Understanding this molecular exchange will help us obtain foundational information regarding the use of microbiota to improve agricultural productivity under drought conditions [84].

### 4.4. Nutrient Deficiency

Nutrient stress, characterized as either suboptimal availability of necessary nutrients or excessive and toxic amounts of essential and non-essential macronutrients, is considered a major stress that has significantly affected crop productivity worldwide. Extracting insoluble nutrients from soils is one of the biggest challenges in the evolution of terrestrial plants; as a result, they may have formed symbiotic relationships with mobile microbes and have flagella or hyphae that can extend into the soil to provide plants with nutrients [125]. Many studies have shown that endophytes can assist plants by transforming and solubilizing nutrients such as N, P, K, and other microminerals, thus making them conveniently accessible to plants [126,127]. For example, *Pseudomonas* sp., as described by Choi et al., mediates phosphate solubilization in rice and wheat by altering gibberellic acid production [128]. Abadi and Sepehri [129] reported that the ability of wheat to absorb mineral nutrients, particularly zinc, was aided by the presence of the endophytes *Azotobacter chroococcum* and *Piriformospora indica*. The bacterial endophyte *Enterobacter* sp. has been reported to help plants adapt to various environmental conditions by improving nutrient acquisition, including nitrogen fixation [130]. Yamaji et al. [131] reported that *Phialocephala fortinii, Rhizodermea veluwensis*, *and Rhizoscyphus* sp. isolated from mining site soil enhanced the growth of *C. barbinervis* seedlings by increasing K uptake in shoots and reducing the concentrations of Cu, Ni, Zn, Cd, and Pb in roots. This suggests that isolating and characterizing endophytes from mining sites may help discover novel endophyte species that can assist in crop improvement under heavy metal stress. Some other examples in Table 2 represent endophytes, their host sand, and their role in various abiotic stresses.

### 4.5. Negative Effects of Endophytes: the Other Side of the Picture

Although endophytes promote the host plants’ growth in several ways [10], these endorsing effects are sensitive to changes in the abiotic environment and may shift from positive to negative when the tolerance range of either symbiont is compromised [147]. The shift in this unstable symbiotic relationship between endophytes and host plants to pathogenicity may be due to variations in abiotic environments, such as nutrient shortages or prolonged extreme weather. We expect this because a fungal species may be endophytic and beneficial to one host species under certain abiotic limits and become pathogenic to the same or another host under different environmental conditions [148]. Despite a good understanding of the behavior and mechanism of endophyte-host plants using high-throughput techniques, there are still substantial gaps regarding microbial lifestyle and their function in host plants [32]. It is therefore inferred that endophytic status cannot be assumed to be certain, as these co-evolved relationships are plastic/dynamic and can be anticipated to destabilize under severe environmental change scenarios [149].

In addition, one should consider the compatibility of endophytes, especially when used as a consortium of endophytes in field conditions. Reports suggested that co-inoculation of chickpea with compatible endophytes such as *Mesorhizobium, Serratia marcescence*, and *Serratia* spp. have synergetic effects on plant growth in terms of nodule number and dry weight, the number of pods per plant, etc. [150]. Similar results were reported by Oliveira et al. [151], where inoculation with a consortium of five diazotrophic bacteria (*Gluconacetobacter diazotrophicus, Herbaspirillum seropedicae, Herbaspirillum rubrisubalbicans, Azospirillum amazonense*, and *Paraburkholderia tropica*) resulted in higher stem production in sugarcane as compared to single endophyte inoculated plants. On the contrary, applying a consortium of incompatible bacteria may have damaging effects on crop growth and production. Co-inoculation of nitrogen-fixing bacteria *Herbaspirillum seropedicae* and *H. rubrisubalbicans* with sugarcane did not increase the overall yield of sugarcane [151]. Therefore, it is highly recommended to do a compatibility test for the endophytes before their application [152].

## 5. Conclusions and Future Recommendations

Our understanding of how to plant endophytes can influence plant physiological responses to biotic and abiotic stresses (see Figure 2). Each year, an increasing number of articles indicate the critical role of plant–microbe interactions in plant fitness during extreme environmental conditions. These breakthroughs have provided new insights into crop protection and disease management. Under unpredictable climates, food crops are more vulnerable to increased biotic and abiotic stresses, putting them at higher risk and jeopardizing the security of the world’s food supply.

Despite significant advancements over the past few decades, there is still a knowledge gap regarding the possibility that environmental stresses can shift plant–microbe interactions from commensal to mutualistic or parasitic relationships. They might compete with each other for survival under adverse conditions, for example, by competing for crucial nutrients such as nitrogen and phosphate [153]. Hence, a better understanding of the factors that determine whether a plant–microbe interaction is harmful or beneficial is required. Transcriptome analysis of perturbed microbial associations in grasses, in which the plant–microbe interaction switches from reciprocal to pathogenic, has identified plant and fungal gene candidates that play a central role [154]. Therefore, we suggest that detailed transcriptomic and metabolomic analyses can be used to comprehensively understand the complex interactions between plants and their microbiome in biotic and abiotic environments. This may provide unprecedented opportunities for enhancing agricultural productivity in extreme environments. Despite the well-established reports on the plant-endophytic symbiotic relationship and the effects on improving plant growth and health, there are no examples of this being used in agriculture. This could be a lack of broadcasting the information from the scientific community to the farmers, a view worth considering.

## Figures and Tables

**Figure 1 cells-11-03292-f001:**
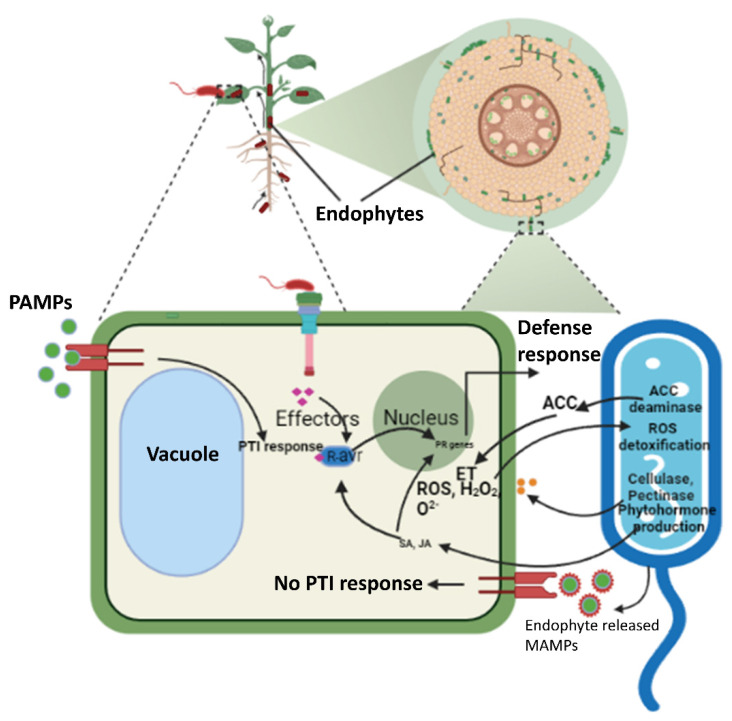
Host plant and endophyte crosstalk. The colonization of endophytes and their interaction with the host plant depends on the type of endophytes. Bacterial endophytes mostly make their entry through apertures formed either through the emergence of lateral roots or the zones of elongation. Fungal endophytes can either be transmitted vertically, i.e., transmission from maternal plants to the progeny seeds, or through their entry as fungal spores through herbivory or insects. Some endophytes also release cell wall–degrading enzymes to make their entry possible. Since these endophytes have specific MAMps, they are not recognized by the host plant PRRs. Endophytes help the plant in stress responses by producing phytohormones and reactive oxygen species (ROS) detoxification. ACC, aminocyclopropane-1-carboxylic acid; ROS, reactive oxygen species; MAMPs, microbe-associated molecular patterns; PR, pathogenesi-related; ET, ethylene; SA, salicylic acid; JA, jasmonic acid; PAMPs, pathogen-associated molecular patterns; PTI, PAMP-triggered immunity; *R*-avr, resistance gene a virulence effector.

**Figure 2 cells-11-03292-f002:**
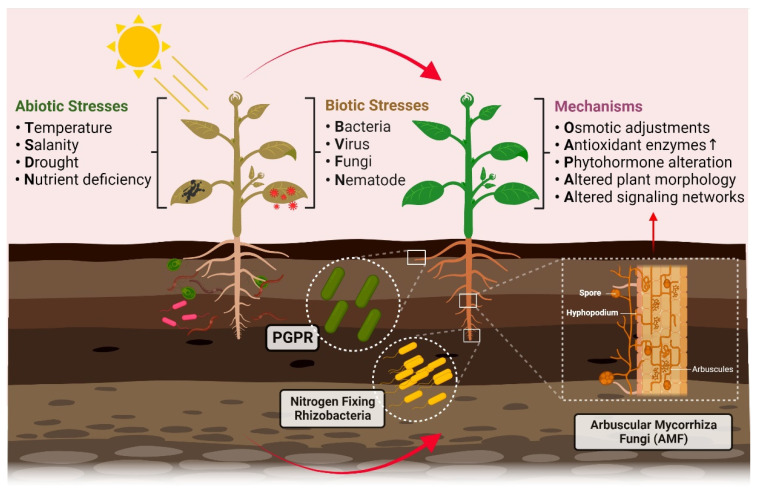
An overview diagram showing major environmental stresses and mechanisms of plant–microbe interactions that influence plant fitness during these stresses. Tolerance against these stresses is mainly mediated by enhancement in antioxidant defense enzymes, osmotic adjustment and production of phytohormones. PGPR: plant growth-promoting rhizobacteria.

**Table 2 cells-11-03292-t002:** List of endophytes mediating abiotic stress tolerance in plants.

Endophyte	Host Plant	Stress	Mechanism	Reference
*Mucilaginibacter* stain K	*Arabidopsis thaliana*	Salinity	Increase in anti-oxidative defense machinery	[132]
*Pantoea agglomerans*	*Zea mays*	Salinity	Upregulation of aquaporins	[133]
*Arthrobacter endophyticus*, *Nocardiopsis alba*	*A. thaliana*	Salinity	Enhancing the expression of genes responsible for water, potassium ion uptake, carotenoid biosynthesis, phenylalanine metabolism, phenylpropanoid biosynthesis, glycerolipid and nitrogen metabolism	[134]
*Penicillium brevicompactum* and *P. chrysogenum*	*Solanum lycopersicum Lactuca sativa*	Salinity	Enhanced energy production and Na^+^ sequestration	[135]
*Alternaria chlamydospora*	*Triticum aestivum*	Salinity	Inducing the physiological and biochemical responses	[136]
*Trichoderma harzianum* TH-56	*Oryza sativa*	Drought	Upregulation of aquaporin, dehydrin, and malonialdehyde genes	[137]
*Pseudomonas putida*	*Cicer arietinum*	Drought	miRNAs and their target genes indicated the involvement in the stress regulatory pathways, which control/regulate drought stress response	[138]
*Bacillus cereus, Pseudomonas otitidis* and *Pseudomonas* sp.	*Glycine max*	Drought	Improving plant growth, membrane integrity, water status, accumulation of compatible solutes, and osmolytes	[139]
*Gluconacetobacter diazotrophicus*	*Saccharum officinarum*	Drought	IAA and proline production	[113]
*Paraphoma* sp., *Embellisia chlamydospora*, and *Cladosporium oxysporum*	Zea mays	Drought	Altered root development	[140]
*Pseudomonas aeruginosa* strain 2CpS1	*T. aestivum*	Heat	Reducing heat and drought–induced oxidative stress	[141]
*Bacillus amyloliquefaciens* subsp. *plantarum* UCMB5113	*T. aestivum*	Heat	by molecular modifications in wheat leaf transcript patterns	[142]
*Bacillus velezensis*	*T. aestivum*	Heat	(1) express several proteins related to stress defense and energy supply (2) modulate several metabolic pathways of amino acids (3) increase the accumulation of GABA in the leaves	[143]
*Burkholderia phytofirmans* PsJN	*A. thaliana*	Cold	Pigment accumulation and cold response pathway induction	[144]
*Bacillus* sp. and *Lysinibacillus* sp.	Argentine screwbean	Nutrient deficiency	Increase in antioxidant defense enzyme, photosynthetic pigments and low lipid peroxidation siderophore, and alteration in IAA, gibberellic acid content	[145]
*Bacillus amyloliquefaciens*	*Oryza sativa*	Nutrient deficiency	Modulates carbohydrate metabolism	[146]
*Pseudomonas* sp.	*Pisum sativum*	Nutrient deficiency	Gluconic acid used by bacteria to solubilize phosphate	[146]

## Data Availability

Not applicable.

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
