# Peer review of "Endophyte-Mediated Stress Tolerance in Plants: A Sustainable Strategy to Enhance Resilience and Assist Crop Improvement"

_cells, 2022, doi:10.3390/cells11203292_

Round 1

Reviewer 1 Report

Manuscript entitled “Endophyte-mediated stress tolerance in plants: a sustainable weapon to enhance resilience and assist crop improvement” by Kamran et al. is a revision of endophyte-mediated stress tolerance in plants.

The authors review different aspects such as colonization and distribution of endophytes, their capacity to modulate plant defense responses, and the interactions between the responses to endophytes and abiotic stresses (emphasizing in the stresses due to temperature, salinity, drought, and nutrient deficiency).

The review is very interesting but, in my opinion, the manuscript presents some issues mainly in the section 2.1 and in the references, and other things, that need to be improved to make this work suitable for publication.

1. Main issues:

1.1. Regarding section “2.1. Colonization and Distribution”

Line 101. Please correct “Non-claviciitaceous” with “Non-clavicitaceous”

Lines 99-106. The authors should address this paragraph because has some inconsistences. According to references 23 the clavicitaceous class 1 has vertical and horizontal transmission. Also it should be pointed out that classes 2, 3 and 4 correspond to Non-clavicitaceous fungal endophytes.

1.2. Regarding References

The references list has numerous format mistakes. The most serious mistakes refer to: many references do not include volume or pages (e.g. reference 6, 13, 38, etc.), many authors do not include the initials of the names (e.g. Roser; Ritchie; Ortiz-Ospina in reference 3, Alix; Capri in reference 6, Sarkar; Ray;…. in reference 38, etc.), many titles include names of organisms that should be italicized (e.g. Vitis vinifera in reference 11, Klebsiella pneumonniae in reference 29, Arabidopsis in reference 35, etc.).

Almost all the words in the title begin with capital letters. Many of them should not be in that format. Please correct this.

In addition to these indicated errors, there are others that must be corrected. I kindly urge to the authors to review the entire format of the references, according to the journal requirements.

2. Additional comments

Line 15. The e-mail address of the author Amna Khatoon is absent

Line 40. Please correct “….by FAO (2018)” with “…by Food and Agriculture Organization (FAO; 2018)”

Line 80. I feel that the sentence “varies from species to species” could be rephrased with “varies across species”

Line 81. The last 9 should be in superscript.

Line 125 and 130. Klebsiella should be italicized

Line 133. Please correct “plant receptors (PRRs)” with “plant pattern recognition receptors (PRRs)”

Line 134. Please correct “their MAMPs” with “their microbe-associated molecular patterns (MAMPs)”.

Lines 135, 137, 219. Please remove the extra parentheses “(“ before the references.

Figure 1. Please correct “Vacule” with “Vacuole”. Also, inside the vacuole there is an additional white oval. Which structure represents?

Line 147. Figure 1 legend. Abbreviations should be described in the legend: PR, R-avr, PTI, PAMPs, ROS, ET, SA, JA, ACC and MAMPs.

Line 153 and 154. Please correct “…SA and JA…” with “…salicylic acid (SA) and jasmonic acid (JA)…”

Line 157. Please add strain before LWL5.

Line 162. The parenthesis and reference “) [47]” should not be in italics.

Line 164. Please correct “..of the ET…” with “… of the ethylene (ET)…”

Line 171. Please add strain before BP25.

Line 177. Please correct “Lichenysin bacillomycin” with “Lichenysin, bacillomycin”

Line 178. What does “,,,” mean?. Please remove excess commas.

Line 178. Please correct “etc [54].” with “etc. [54].”

Figure 2. In this figure and line 191 Heavy metal stress is mencioned, however the authors in the manuscript do not develop the discussion of this kind of stress, except from a brief mention in line 371. This stress should be addressed in a major extent.

Line 202. Figure 2 legend. Please add the meaning of the abbreviation PGPR, plant growth-promoting rhizobacteria.

Line 208. Please correct “1.1-5.4C” with “1.1-5.4 ºC”

Lines 211, 212, 223,226, 246, 251, 256, 262, 269, 276, 280, 284, 291, 297, 305, 307, 315, 316, 320, 335, 336, 358, 391. In all cases a space is missing before the references.

Line 214. Please add strain before PsJN.

Line 229. Please add strain before SA1.

Line 232. ”strain AKMP7” should not be in italics.

Line 233. ”strain AKMP6” should not be in italics.

Line 233. In the sentence there is an extra space between the words “and Pseudomonas”

Line 232. ”sp. strain” should not be in italics.

Line 239. “C. protuberate” should be italicized

Line 246. In the sentence there is an extra space between the words “systems, salt”

Line 262. In the sentence there is an extra space between the words “proline, glycine”.

Line 265. In the sentence there is an extra space between the words “tabaci, Curtobacte-”.

Line 268. Please correct “3-acetic acid abscisic acid” with “3-acetic acid, abscisic acid (ABA)”

Line 274 y 283. The “+” should be in superscript.

Line 277. In the sentence there is an extra comma between the words “Bacillus amyloliquefaciens

Line 284. I feel that the sentence “proposed that inoculated seedlings' roots had….” could be rephrased with “proposed that roots from inoculated seedlings had….”

Line 306. Please correct “reduced, reactive oxygen species (ROS) are produced” with “reduced, ROS are produced”.

Line 313. Please add strain before DE10.

Line 316. Reference “[101]” should not be in italics.

Line 317. The full stop at the end of the sentence is missing.

Line 319. In the sentence there is an extra space between the words “instances, Rhizophagus”.

Line 320. In the sentence there is an extra dot between the words “roots [105]”.

Line 322. Please correct “AM fungi” with “arbuscular mycorrhiza fungi”.

Line 334. Bacillus should be italicized.

Line 340. In the sentence there is an extra space between the words “et al. [114]”.

Line 361. Please correct “pseudomonas sp” with “Pseudomonas sp”.

Line 362. In the sentence there is an extra space between the words “Sepehri [119]”.

Line 367. In the sentence there is an extra dot between the words “[121] reported”.

Line 367. Please correct “Rhizoscyphus sp. Isolated“ with “Rhizoscyphus sp. isolated“.

Author Response

Responses to Reviewer 1 comments

Manuscript entitled “Endophyte-mediated stress tolerance in plants: a sustainable weapon to enhance resilience and assist crop improvement” by Kamran et al. is a revision of endophyte-mediated stress tolerance in plants.

The authors review different aspects such as colonization and distribution of endophytes, their capacity to modulate plant defense responses, and the interactions between the responses to endophytes and abiotic stresses (emphasizing in the stresses due to temperature, salinity, drought, and nutrient deficiency).

The review is very interesting but, in my opinion, the manuscript presents some issues mainly in the section 2.1 and in the references, and other things, that need to be improved to make this work suitable for publication.

 Answer.  We are thankful for your time to improve the manuscript. Please see below our answers to your questions, with a point-by-point reply. Many of the English grammar and typos mistakes were already corrected by the English editing service, others we corrected as per your suggestions. Sorry for the inconvenience caused. We have highlighted the changes with the green color

  1. Main issues:

1.1. Regarding section “2.1. Colonization and Distribution”

Line 101. Please correct “Non-claviciitaceous” with “Non-clavicitaceous”

Lines 99-106. The authors should address this paragraph because has some inconsistences. According to references 23 the clavicitaceous class 1 has vertical and horizontal transmission. Also, it should be pointed out that classes 2, 3 and 4 correspond to Non-clavicitaceous fungal endophytes.

Answer: Thank you for the thorough revision and the time given to the manuscript. We have now addressed these issues in the revised manuscript and the changes are highlighted green.

1.2. Regarding References

The references list has numerous format mistakes. The most serious mistakes refer to: many references do not include volume or pages (e.g. reference 6, 13, 38, etc.), and many authors do not include the initials of the names (e.g. Roser; Ritchie; Ortiz-Ospina in reference 3, Alix; Capri in reference 6, Sarkar; Ray;…. in reference 38, etc.), many titles include names of organisms that should be italicized (e.g. Vitis vinifera in reference 11, Klebsiella pneumonniae in reference 29, Arabidopsis in reference 35, etc.).

Almost all the words in the title begin with capital letters. Many of them should not be in that format. Please correct this.

In addition to these indicated errors, there are others that must be corrected. I kindly urge to the authors to review the entire format of the references, according to the journal requirements.

Answer: The major problem with reference formatting was due to the fact that different authors were using different referencing software, which caused troubles, we have now revised the whole manuscript and hope that they are fine now.

  1. Additional comments

Line 15. The e-mail address of the author Amna Khatoon is absent

Answer. The email address of the author is now added to the revised manuscript. Thanks for pointing out

Line 40. Please correct “….by FAO (2018)” with “…by Food and Agriculture Organization (FAO; 2018)”

 Answer. Changed as per reviewer’s suggestion

Line 80. I feel that the sentence “varies from species to species” could be rephrased with “varies across species”

 Answer. Changed as per reviewer’s suggestion in the revised manuscript.

Line 81. The last 9 should be in superscript.

 Answer. Changed as per reviewer’s suggestion in the revised manuscript.

Line 125 and 130. Klebsiella should be italicized

 Answer. Changed as per reviewer’s suggestion, now line 124 and 129 in the revised manuscript.

Line 133. Please correct “plant receptors (PRRs)” with “plant pattern recognition receptors (PRRs)”

 Answer. Changed as per reviewer’s suggestion now line 132-133 in the revised manuscript.

Line 134. Please correct “their MAMPs” with “their microbe-associated molecular patterns (MAMPs)”.

 Answer. Changed as per reviewer’s suggestion in the revised manuscript.

Lines 135, 137, 219. Please remove the extra parentheses “(“ before the references.

 Answer. Extra parentheses were removed as per the reviewer’s suggestion in the revised manuscript.

Figure 1. Please correct “Vacule” with “Vacuole”. Also, inside the vacuole there is an additional white oval. Which structure represents?

Answer. Figure 1 is now modified as per the reviewer’s suggestion, the extra oval was part of the vacuole we have now re-drawn this part to avoid confusion.

Line 147. Figure 1 legend. Abbreviations should be described in the legend: PR, R-avr, PTI, PAMPs, ROS, ET, SA, JA, ACC and MAMPs.

Answer. Thanks for pointing out this, we have now added abbreviations in the revised manuscript Figure 1 caption.

 Line 153 and 154. Please correct “…SA and JA…” with “…salicylic acid (SA) and jasmonic acid (JA)…”

Answer. Changed as per reviewer’s suggestion in the revised manuscript now line number 157.

Line 157. Please add strain before LWL5.

Answer. Changed as per reviewer’s suggestion in the revised manuscript.

Line 162. The parenthesis and reference “) [47]” should not be in italics.

Answer. They are now fixed in the revised manuscript now reference number 39 line 167.

Line 164. Please correct “..of the ET…” with “… of the ethylene (ET)…”

Answer. The full name for ET is now added in the revised manuscript now line 169

Line 171. Please add strain before BP25.

Answer. This is now added to the revised manuscript

Line 177. Please correct “Lichenysin bacillomycin” with “Lichenysin, bacillomycin”

Answer. A comma is added as per the reviewer’s suggestion in the revised manuscript.

Line 178. What does “,,,” mean?. Please remove excess commas.

Answer. It was a typo mistake, we have now corrected it in the revised manuscript.

Line 178. Please correct “etc [54].” with “etc. [54].”

Answer. This part is now removed by the English editing service in the revised manuscript.

Figure 2. In this figure and line 191 Heavy metal stress is mencioned, however the authors in the manuscript do not develop the discussion of this kind of stress, except from a brief mention in line 371. This stress should be addressed in a major extent.

Answer.  Thank you for highlighting this. There are many abiotic stresses, but our intention was only to discuss four major stresses as examples, such as temperature, drought salinity, and nutrient stress. Including further stresses will inflate the manuscript. However, to avoid confusion, we have made corrections in line 191 and figure 2 and removed the heavy metal part from there to be consistent.

Line 202. Figure 2 legend. Please add the meaning of the abbreviation PGPR, plant growth-promoting rhizobacteria.

 Answer. We have added the abbreviation in the revised manuscript. Thanks for mentioning it.

Line 208. Please correct “1.1-5.4C” with “1.1-5.4 ºC”

Answer. This is corrected in the revised manuscript now line 220.

Lines 211, 212, 223,226, 246, 251, 256, 262, 269, 276, 280, 284, 291, 297, 305, 307, 315, 316, 320, 335, 336, 358, 391. In all cases a space is missing before the references.

Answer. This has been fixed in the revised manuscript.

Line 214. Please add strain before PsJN

Answer. Added as suggested.

Line 229. Please add strain before SA1.

Answer. Added as suggested.

Line 232. ”strain AKMP7” should not be in italics.

Answer. This is now corrected in the revised manuscript line 244.

Line 233. ”strain AKMP6” should not be in italics.

Answer.  Fixed as suggested by the reviewer.

Line 233. In the sentence there is an extra space between the words “and Pseudomonas”

Answer.  Fixed as suggested by the reviewer.

Line 232. ”sp. strain” should not be in italics.

Answer.  Fixed as suggested by the reviewer.

Line 239. “C. protuberate” should be italicized

Answer.  Fixed as suggested by the reviewer.

Line 246. In the sentence there is an extra space between the words “systems, salt”

Answer.  Fixed as suggested by the reviewer.

Line 262. In the sentence there is an extra space between the words “proline, glycine”.

Answer.  Fixed as suggested by the reviewer.

Line 265. In the sentence there is an extra space between the words “tabaci, Curtobacte-”.

Answer.  Fixed as suggested by the reviewer.

Line 268. Please correct “3-acetic acid abscisic acid” with “3-acetic acid, abscisic acid (ABA)”

Answer.  The term abscisic acid is now abbreviated in the revised manuscript.

Line 274 y 283. The “+” should be in superscript.

Answer.  Fixed as suggested by the reviewer.

Line 277. In the sentence there is an extra comma between the words “Bacillus amyloliquefaciens

Answer.  Extra comma removed in the revised manuscript

Line 284. I feel that the sentence “proposed that inoculated seedlings' roots had….” could be rephrased with “proposed that roots from inoculated seedlings had….”

Answer.  The sentence is now revised now lines 296-297.

Line 306. Please correct “reduced, reactive oxygen species (ROS) are produced” with “reduced, ROS are produced”.

Answer.  This is now changed in the revised manuscript line number 318.

Line 313. Please add strain before DE10.

Answer.  Fixed as suggested by the reviewer.

Line 316. Reference “[101]” should not be in italics.

Answer.  Fixed as suggested by the reviewer.

Line 317. The full stop at the end of the sentence is missing.

Answer.  Fixed as suggested by the reviewer.  Now line 329

Line 319. In the sentence there is an extra space between the words “instances, Rhizophagus”.

Answer.  Fixed as suggested by the reviewer.

Line 320. In the sentence there is an extra dot between the words “roots [105]”.

Answer.  Fixed as suggested by the reviewer.

Line 322. Please correct “AM fungi” with “arbuscular mycorrhiza fungi”.

Answer.  This is now corrected in the revised manuscript line 334.

Line 334. Bacillus should be italicized.

Answer.  Corrected as suggested by the reviewer, now line 346

Line 340. In the sentence there is an extra space between the words “et al. [114]”.

Answer.  Corrected as suggested by the reviewer.

Line 361. Please correct “pseudomonas sp” with “Pseudomonas sp”.

Answer.  Corrected as suggested by the reviewer. Now line 373

Line 362. In the sentence there is an extra space between the words “Sepehri [119]”.

Answer.  Corrected as suggested by the reviewer.

Line 367. In the sentence there is an extra dot between the words “[121] reported”.

Answer.  Corrected as suggested by the reviewer.

Line 367. Please correct “Rhizoscyphus sp. Isolated“ with “Rhizoscyphus sp. isolated“.

 Answer.  Corrected as suggested by the reviewer. Now line 381

Reviewer 2 Report

In this article Kamran et al. have reported about endophytes and their role in mediating plant stress responses. The authors explained about the colonization of endophytes and how they modulate plant defense responses during colonization. They further mentioned the role of endophytes in resistance against phytopathogens and abiotic factors such as temperature, salinity and drought etc. The manuscript presents interesting information and could be a good piece of literature for understanding plant-microbe interaction.

I have a few comments/suggestions.

Abstract: In the introduction section, the authors mentioned the use of various agrochemicals that might have potential health risks and that endophytes could be a better alternative for sustainable agriculture, I would suggest summarizing it in the abstract too.

 In section1. Major challenges…..production, the author focused mainly on the use of agrochemicals etc., which does not seem to be a problem for the plant but rather for humans, the author should discuss here biotic and abiotic factors briefly that are threats to plant production.

 It would be good to include a table or two including endophytes that play a role in biotic or abiotic stresses.

 Minor comments:

Line 81: 9 x 109 “9” should be in superscript.

Line 263: replace  reprograming with “reprogramming”

 In Figure 2 Abiotic stress should be “Abiotic Stresses” and Biotic Stress should be “Biotic Stresses”

Figure 2 should be described in detail, a figure should be self-explanatory

Author Response

Responses to Reviewer 2 comments

In this article Kamran et al. have reported about endophytes and their role in mediating plant stress responses. The authors explained about the colonization of endophytes and how they modulate plant defense responses during colonization. They further mentioned the role of endophytes in resistance against phytopathogens and abiotic factors such as temperature, salinity, and drought etc. The manuscript presents interesting information and could be a good piece of literature for understanding plant-microbe interaction.

I have a few comments/suggestions.

Abstract: In the introduction section, the authors mentioned the use of various agrochemicals that might have potential health risks and that endophytes could be a better alternative for sustainable agriculture, I would suggest summarizing it in the abstract too.

Answer.  Thank you for raising this point. We have now made some changes in the abstract in line with the reviewer’s suggestions.

 In section1. Major challenges…..production, the author focused mainly on the use of agrochemicals etc., which does not seem to be a problem for the plant but rather for humans, the author should discuss here biotic and abiotic factors briefly that are threats to plant production.

Answer.  The ultimate purpose of improving plant health and fitness is to serve human needs. If we have problems with plant growth, there will be shortage, or nutrient deficient food indirectly affecting human health.

 It would be good to include a table or two including endophytes that play a role in biotic or abiotic stresses.N

Answer.  We have now added two tables each for endophytic role in biotic and abiotic stresses.

 Minor comments:

Line 81: 9 x 109 “9” should be in superscript.

Answer.  This is now corrected in the revised manuscript

Line 263: replace  reprograming with “reprogramming”

Answer.  This is now corrected in the revised manuscript

 In Figure 2 Abiotic stress should be “Abiotic Stresses” and Biotic Stress should be “Biotic Stresses”

Answer.  This is now corrected in the revised manuscript

Figure 2 should be described in detail, a figure should be self-explanatoryM

Answer.  A detailed figure legend is now added to Figure 2

Reviewer 3 Report

My review of the manuscript of “Endophyte-mediated stress tolerance in plants: a sustainable 2 weapon to enhance resilience and assist crop improvement (Manuscript ID: cells-1900914) is as follows:

Agriculture is extremely exposed to weather extremites. As a result of climate change, the average temperature of our Earth increases and the frequency of extreme weather conditions, including those of drought years. This review article presents the importance of endophytic microorganisms in the stress tolerance of crops (by both abiotic- and biotic stresses).

The formal style of this review article and the quality of the figures are appropriate. The manuscript is based on 123 articles, about 24% of which were written in the last 5 years. In my opinion, better use could have been made of the cited articles. As the topic is current, it would also justify the inclusion of a larger proportion of recent publications. The language of the manuscript is understandable, but there are many small errors in the text, which are extremely confusing (typos, incorrect description of the scientific name of species, etc.). Interestingly, the scientific names of the plant species were not listed by the authors (or only rarely and irregularly).

The four endophytic classes mentioned in the introduction have no role in the rest of the manuscript. So it's not clear to me why it was included at all. The entire manuscript discusses the positive effect of fungal and bacterial endosymbionts. However, there are significant differences between their mechanisms of action. Such a generalization and amalgamation of these is not a good idea. Furthermore, I am missing one table (or, based on the above, even two) in which it would have been possible to summarize which endophyte and host plant species have been studied under which stress and which genes are activated/repressed (in relation to endophyte and host).

In terms of content, unfortunately, I do not feel that the manuscript would add anything to our existing knowledge of endophytes related to plant stress tolerance. A deeper understanding of the mechanisms of stress tolerance induced by endophytes and the conclusions thereof are lacking. Furthermore, unfortunately, I do not feel that the manuscript provides a management-, application approaches for practicioners.

For the reasons mentioned above, I unfortunately have to reject the current form of the manuscript.

Author Response

Responses to Reviewer 3 comments

My review of the manuscript of “Endophyte-mediated stress tolerance in plants: a sustainable 2 weapon to enhance resilience and assist crop improvement” (Manuscript ID: cells-1900914) is as follows:

Agriculture is extremely exposed to weather extremites. As a result of climate change, the average temperature of our Earth increases and the frequency of extreme weather conditions, including those of drought years. This review article presents the importance of endophytic microorganisms in the stress tolerance of crops (by both abiotic- and biotic stresses).

The formal style of this review article and the quality of the figures are appropriate. The manuscript is based on 123 articles, about 24% of which were written in the last 5 years. In my opinion, better use could have been made of the cited articles. As the topic is current, it would also justify the inclusion of a larger proportion of recent publications. The language of the manuscript is understandable, but there are many small errors in the text, which are extremely confusing (typos, incorrect description of the scientific name of species, etc.). Interestingly, the scientific names of the plant species were not listed by the authors (or only rarely and irregularly).

Answer: Thank you for your time and valuable suggestions to improve the manuscript. All the suggestions were helpful. We have tried our best to collect recent articles on the topic with the consideration that the pioneer workers are not ignored and properly acknowledged. We are sorry for the inconvenience caused due to typo mistakes and syntax problems, we, therefore, revised the paper from an official English Editing service and hope that language issues are now resolved. Some of the common plants were not used with scientific nomenclature, however, we have added tables in response to your other comments in which we have used scientific names for the plants.

The four endophytic classes mentioned in the introduction have no role in the rest of the manuscript. So it's not clear to me why it was included at all. The entire manuscript discusses the positive effect of fungal and bacterial endosymbionts. However, there are significant differences between their mechanisms of action. Such a generalization and amalgamation of these is not a good idea. Furthermore, I am missing one table (or, based on the above, even two) in which it would have been possible to summarize which endophyte and host plant species have been studied under which stress and which gene

Answer: The four classes mentioned in the introduction were mere for introducing fungal endophytes and their major classes for the common reader. Therefore, familiarizing readers with the major classification of Calviipitaceous and Noncalvicipitaceous endophytes would be beneficial from our point of view. We agree with the reviewer, that instinctively the paper was more towards the positive roles of endophytes which was also the intended objective of this manuscript, however, to give the reader both perspectives, we have now added one brief section about “Negative Effects of Endophytes, the other side of the picture”. We have now added two tables each for the role of endophytes in biotic and abiotic stresses at the end of the revised manuscript as per the reviewers’ suggestion.

In terms of content, unfortunately, I do not feel that the manuscript would add anything to our existing knowledge of endophytes related to plant stress tolerance. A deeper understanding of the mechanisms of stress tolerance induced by endophytes and the conclusions thereof are lacking. Furthermore, unfortunately, I do not feel that the manuscript provides a management-, application approaches for practicioners.

For the reasons mentioned above, I unfortunately have to reject the current form of the manuscript.

Answer: We are sorry that the manuscript was not up to the expectations. Agreeing with the reviewer that the topic is current and growing. New endophytes with diverse roles are exploring day by day each with a different way of mitigating stress, therefore, it is very difficult to rely on one mechanism. We therefore in our concluding section mentioned that detailed transcriptomic and metabolomics approaches are needed to explore the underlying mechanisms.

Reviewer 4 Report

The article presents an interestingly well-known relationship between plants and endophytes in the context of resistance to biotic and abiotic factors. However, there were no examples of symbiotic plant-endophyte systems already implemented in agricultural practice, which could be a very valuable complement.

Below, please find some remarks and corrections to the text:

Line # 21: But in fact we use 'avoidance' and 'escape' terms to describe plant strategies against drought?

Line # 23: ‘Mobile companions’ is a rather exaggerated term - endophytes are mobile the same way as plants are….

Line # 81:  9 x 109 - 10 to the 9th power.

Line # 178: ‘, , , , “ – something missing?

Line # 330: Check the correct citation for [108]

Line # 357: ‘mobile’ instead of ‘motile’?

Line # 361: Pseudomonas

Author Response

Reviewer 4

Comments and Suggestions for Authors

The article presents an interestingly well-known relationship between plants and endophytes in the context of resistance to biotic and abiotic factors. However, there were no examples of symbiotic plant-endophyte systems already implemented in agricultural practice, which could be a very valuable complement.

Answer: Thank you for your time and valuable suggestions. We agree with the reviewer that there are no reported examples (to our knowledge) of the application of plant-endophyte in the agriculture system. We believe this could be due to a lack of knowledge transfer from the scientific community to the ordinary farmers. We have added this in our conclusion and future recommendations section.

Below, please find some remarks and corrections to the text:

Line # 21: But in fact we use 'avoidance' and 'escape' terms to describe plant strategies against drought?

Answer: We are sorry for the inconvenience, we have now corrected the sentence to avoid confusion in the revised manuscript, now lines 24-25.

Line # 23: ‘Mobile companions’ is a rather exaggerated term - endophytes are mobile the same way as plants are….

Answer: We have removed “mobile companions” from the sentence for more clarity.

Line # 81:  9 x 10- 10 to the 9th power.

Answer: This has now been corrected in the revised manuscript.

Line # 178: ‘, , , , “ – something missing?

Answer: This was a typo and was fixed in the revised version

Line # 330: Check the correct citation for [108]

Answer: Thank you for pointing out this, we have now corrected the citation in the revised manuscript now line 348.

Line # 357: ‘mobile’ instead of ‘motile’?

Answer: The suggested changes are made to the revised manuscript, now line 374.

Line # 361: Pseudomonas

This has been corrected now in the revised version now line 377.

Round 2

Reviewer 1 Report

The authors have made a considerable improvement in the manuscript but there are still things to modify.

 Figure 1. Please correct “Vacoule” with “Vacuole”.

Line 149 to 152. Figure 1 legend. PR abbreviation should be described in the legend.

Line 162. Please add strain before LWL5.

Line 219. Please correct “5.4 ºC” with “5.4ºC”. Remember that the units should appear separated from the numbers, with the exception of ºC, and %.

Line 244, 378. ”sp.” should not be in italics

Line 251. Please check de references. Reference 73 between 64 and 65

Line 344. Please check “enzimes-[99]”

Line 383. Please correct “Table. 2” with “Table 2”

The authors should put more attention with the reference format

References 7, 9, 10, 12, 14, 17, 20, 26, 27, 28, 29, 33, 34, 36, 37, 39, 42?, 45, 48, 49, 50, 51 to 54, 57, 59,  61, 62, 63, 65?, 69, 72, 74, 75, 79, 80, 81, 82, 83, 84, 88, 89, 90, 91, 96, 98, 99, 102, 102, 107, 109, 111, 113, 115, 116, 117, 118, 123, 128, 129, 137, 138, 143, 144, and 145. The journal name should be abbreviated and in italics

Reference 64. Please correct “science” with “Science”.

Reference 17, 47, 49, 91, 104, 123, 127. These references do not include volume or pages

Reference 11, 15, 16, 19, 21, 22, 25, 30, 31, 32, 35, 38, 40, 41, 43, 44, 46, 47, 55, 56, 58, 60, 67, 68, 70, 73, 76, 77, 78, 85, 86, 87, 92, 93, 94, 95, 97, 100, 105, 108, 112, 119, 120, 112, 122, 123, 124, 126, 127, 131, 132, 133, 134, 135, 136, 139, 140, 141, 142, 147, 148, 149, 150, 151. Please, remove the DOI

Please check the references thoroughly

Table 1 and 2. Please check the names. Genus and species in italics (e.g. Bacillus in row 6, Fusarium solani, Zea mays. ”sp.” and “spp.” should not be in italics. In “Helianthus annus L” the “L” should not be in italics

Author Response

Responses to Reviewer 1 comments

The authors have made a considerable improvement in the manuscript but there are still things to modify.

Answer: Thank you for time and consideration. We have tried to figure out typo-mistakes and corrected them as per reviewer’s suggestions. References styles may change when we add or remove references based on reviewer’s suggestions. We, therefore, normally check them once the paper is finalized. We have revised the manuscript before submission, and hope that the typos are addressed, if needed, final formatting can be done at proof reading stage also.

 Figure 1. Please correct “Vacoule” with “Vacuole”.

 Answer.  Thank you for pointing out this mistake, we have now corrected it in the revised version

Line 149 to 152. Figure 1 legend. PR abbreviation should be described in the legend.

Answer.  PR abbreviation is now described in the revised version

Line 162. Please add strain before LWL5.

Answer.  Added as suggested

Line 219. Please correct “5.4 ºC” with “5.4ºC”. Remember that the units should appear separated from the numbers, with the exception of ºC, and %.

Answer.  Corrected as per suggestion

Line 244, 378. ”sp.” should not be in italics

Answer.  Thank you for pointing this, we have now corrected it

Line 251. Please check de references. Reference 73 between 64 and 65

Answer.  We have now corrected it

Line 344. Please check “enzimes-[99]”

Answer.  Corrected as suggested

Line 383. Please correct “Table. 2” with “Table 2”

 Answer.  Corrected as suggested

The authors should put more attention with the reference format

References 7, 9, 10, 12, 14, 17, 20, 26, 27, 28, 29, 33, 34, 36, 37, 39, 42?, 45, 48, 49, 50, 51 to 54, 57, 59,  61, 62, 63, 65?, 69, 72, 74, 75, 79, 80, 81, 82, 83, 84, 88, 89, 90, 91, 96, 98, 99, 102, 102, 107, 109, 111, 113, 115, 116, 117, 118, 123, 128, 129, 137, 138, 143, 144, and 145. The journal name should be abbreviated and in italics

Answer.  The journal names were abbreviated, but as I said, when we update reference, the changes are reversed. Therefore, it would be good if these changes are made once the paper is accepted and there are no further scientific questions to be answered, then one can focus on formatting issues

 Reference 64. Please correct “science” with “Science”.

Answer.  Corrected as suggested

Reference 17, 47, 49, 91, 104, 123, 127. These references do not include volume or pages

Reference 11, 15, 16, 19, 21, 22, 25, 30, 31, 32, 35, 38, 40, 41, 43, 44, 46, 47, 55, 56, 58, 60, 67, 68, 70, 73, 76, 77, 78, 85, 86, 87, 92, 93, 94, 95, 97, 100, 105, 108, 112, 119, 120, 112, 122, 123, 124, 126, 127, 131, 132, 133, 134, 135, 136, 139, 140, 141, 142, 147, 148, 149, 150, 151. Please, remove the DOI

Please check the references thoroughly

Answer.  Thank you very much for detail overview of the manuscript. We have tried our best to focus on formatting now. As suggested in the author guidelines, we have used Endnote template for references. There might be typo mistakes that we can deal with once everything is finalized since the formatting is changed back when a new reference is added or removed.

Table 1 and 2. Please check the names. Genus and species in italics (e.g. Bacillus in row 6, Fusarium solani, Zea mays. ”sp.” and “spp.” should not be in italics. In “Helianthus annus L” the “L” should not be in italics

Answer.  Thank you for pointing out this, we have now corrected it in the revised manuscript.

Reviewer 3 Report

My second review of the manuscript of “Endophyte-mediated stress tolerance in plants: a sustainable 2 weapon to enhance resilience and assist crop improvement” (Manuscript ID: cells-1900914) is as follows:

I find the manuscript suitable for publication. The requested corrections were made by the author, for which I thank them. I would just like to draw your attention to a few minor things.

Line 384: (4.5. Negative Effects of Endophytes, the other side of the picture)

Can we talk about an imcompatibility between certain plant and endophyte species? It can be interesting from the endophyte applicability point of view.

L 43: the reference should be numbered

L 172: Move Table 1 here. (figures must be placed after their first appearance in the text).

L 262, 269, 280, 311, 318, 338-339, 349, 408: empty line, please delete it.

L 378: The word 'and' is not italicized “..Rhizodermea veluwensis, and Rhizoscyphus sp..”.

L 384: Move Table 2 here. (figures must be placed after their first appearance in the text).

L 434: Conflict of interest statement are missing. Please fill in correctly.

Table 1, Ref. each of 119: A dot is missing (Bacillus spp.) (Column of Enophyte).

Table 1, Ref. 120, 125 and 128: Missing ‘ssp.’ (Paenibacillus, Paraburkholderia and Flavobacterium) (Column of Enophyte).

Table 1, Ref. 121: Target pathway are consistently capitalized. “Antioxidant” instead of “antioxidant”.

Table 1, Ref. 124: Double space before the 'Sclerotium' (Column of Pathogen).

Table 1, Ref. 125 and 35: Scientific names of species are written in italics (Paraburkholderia, Fusarium solani) (Column of Enophyte).

Table 1, Ref. 127, 128: A dot is missing (Column of Enophyte).

Table 1, Ref. 135: the word 'syringae' is written in lower case (Column of Pathogen).

Table 2, Ref. 140: The word 'and' is not italicized (Column of Enophyte). A comma is missing (Column of Host plant).

Table 2, Ref. 144: Double space before the word 'and' (Column of Enophyte). Stress forms are consistently capitalized. “Drought” instead of “drought”.

Table 2, Ref. 145: Stress forms are consistently capitalized. “Drought” instead of “drought”.

Table 2, Ref. 146: Incorrect font (Column of Enophyte).

Table 2, Ref. 148: Delete the dot from the species name (Column of Enophyte).

Author Response

Responses to Reviewer 3 comments

Reviewer 3

My second review of the manuscript of “Endophyte-mediated stress tolerance in plants: a sustainable 2 weapon to enhance resilience and assist crop improvement” (Manuscript ID: cells-1900914) is as follows:

I find the manuscript suitable for publication. The requested corrections were made by the author, for which I thank them. I would just like to draw your attention to a few minor things.

Answer: Thank you so much for your valuable time and constructive suggestions. We appreciate revision that improve the overall impact of the manuscript. All the suggestions were really good, and we have tried our best to address those in our revised manuscript. I highlighted them in yellow

 Line 384: (4.5. Negative Effects of Endophytes, the other side of the picture)

Can we talk about an imcompatibility between certain plant and endophyte species? It can be interesting from the endophyte applicability point of view.

Answer: Yes, it is a good suggestion, we have now added following paragraph about compatibility in the revised manuscript line 391-404.

In addition, one should consider the compatibility of endophytes, especially when used as a consortium of endophytes in field conditions. Reports suggested that co-inoculation of chickpea with compatible endophytes such as Mesorhizobium, Serratia marcescence, and Serratia spp have synergetic effects on plant growth in terms of nodule number and dry weight, and the number of pods per plant, etc. [1]. Similar results were reported by Oliveira, et al. [2] where inoculation with a consortium of five diazotrophic bacteria  Gluconacetobacter  diazotrophicus, Herbaspirillum seropedicae, Herbaspirillum rubrisubalbicans, Azospirillum  amazonense, and Paraburkholderia tropica) resulted in higher stem production in sugarcane as compared to single endophyte inoculated plants. On the contrary, applying a consortium of incompatible bacteria may have damaging effects on crop growth and production. Co-inoculation of nitrogen-fixing bacteria Herbaspirillum seropedicae and H. rubrisubalbicans with sugarcane did not increase the overall yield of sugarcane [2] Therefore, it is highly recommended to do a compatibility test for the endophytes before their application [3].

L 43: the reference should be numbered

Answer: The reference is now corrected as per reviewer’s suggestion.

L 172: Move Table 1 here. (figures must be placed after their first appearance in the text).

Answer: Table 1 is now moved to appropriate position in the revised manuscript

L 262, 269, 280, 311, 318, 338-339, 349, 408: empty line, please delete it.

Answer: Empty lines have now been deleted

L 378: The word 'and' is not italicized “..Rhizodermea veluwensisand Rhizoscyphus sp..”.

Answer: Thank you for pointing, we have now corrected this in the revised manuscript.

L 384: Move Table 2 here. (figures must be placed after their first appearance in the text).

Answer: Table 2 is now moved to appropriate position in the revised manuscript

L 434: Conflict of interest statement are missing. Please fill in correctly.

Answer: We have now added conflict of interest statement in the revised manuscript

Table 1, Ref. each of 119: A dot is missing (Bacillus spp.) (Column of Enophyte).

Answer: We have fixed it in the revised manuscript.

Table 1, Ref. 120, 125 and 128: Missing ‘ssp.’ (Paenibacillus, Paraburkholderia and Flavobacterium) (Column of Enophyte).

Answer: We have fixed it in the revised manuscript

Table 1, Ref. 121: Target pathway are consistently capitalized. “Antioxidant” instead of “antioxidant”.

Answer: Thank you for pointing out this, we have now corrected it in the revised manuscript.

Table 1, Ref. 124: Double space before the 'Sclerotium' (Column of Pathogen).

Answer: We have fixed it in the revised manuscript

Table 1, Ref. 125 and 35: Scientific names of species are written in italics (Paraburkholderia, Fusarium solani) (Column of Enophyte).

Answer: We have fixed it in the revised manuscript

Table 1, Ref. 127, 128: A dot is missing (Column of Enophyte).

Answer: We have fixed it in the revised manuscript

Table 1, Ref. 135: the word 'syringae' is written in lower case (Column of Pathogen).

Answer: We have corrected it in the revised manuscript

 Table 2, Ref. 140: The word 'and' is not italicized (Column of Enophyte). A comma is missing (Column of Host plant).

Answer: We have corrected it as per suggestions

Table 2, Ref. 144: Double space before the word 'and' (Column of Enophyte). Stress forms are consistently capitalized. “Drought” instead of “drought”.

Answer: We have fixed it in the revised manuscript

Table 2, Ref. 145: Stress forms are consistently capitalized. “Drought” instead of “drought”.

Table 2, Ref. 146: Incorrect font (Column of Enophyte).

Answer: We have fixed it in the revised manuscript

Table 2, Ref. 148: Delete the dot from the species name (Column of Enophyte).

Answer: We have fixed it in the revised manuscript

  1. Shahzad, S.M.; Khalid, A.; Arif, M.S.; Riaz, M.; Ashraf, M.; Iqbal, Z.; Yasmeen, T. Co-inoculation integrated with P-enriched compost improved nodulation and growth of Chickpea (Cicer arietinum L.) under irrigated and rainfed farming systems. Biology and fertility of soils 2014, 50, 1-12.
  2. Oliveira, A.; Stoffels, M.; Schmid, M.; Reis, V.; Baldani, J.; Hartmann, A. Colonization of sugarcane plantlets by mixed inoculations with diazotrophic bacteria. european journal of soil biology 2009, 45, 106-113.
  3. Batra, P.; Barkodia, M.; Ahlawat, U.; Sansanwal, R.; Wati, L. Effect of Compatible and Incompatible Endophytic Bacteria on Growth of Chickpea Plant. Def. Life Sci. J 2020, 5, 45-48.